# Biomechanical Analysis of Unplanned Gait Termination According to a Stop-Signal Task Performance: A Preliminary Study

**DOI:** 10.3390/brainsci13020304

**Published:** 2023-02-10

**Authors:** Dong-Kyun Koo, Jung-Won Kwon

**Affiliations:** 1Department of Public Health Sciences, Graduate School, Dankook University, Cheonan 31116, Republic of Korea; 2Department of Physical Therapy, College of Health and Welfare Sciences, Dankook University, Cheonan 31116, Republic of Korea

**Keywords:** gait termination, 3D motion capture, response inhibition

## Abstract

There is a correlation between cognitive inhibition and compensatory balance response; however, the correlation between response inhibition and gait termination is not clear. Objectives: The purpose of this study was to investigate the gait parameters of the lower extremity that occurred during unplanned gait termination (UGT) in two groups classified by the stop-signal reaction time (SSRT). Methods: Twenty young adults performed a stop-signal task and an unplanned gait termination separately. UGT required subjects to stop on hearing an auditory cue during randomly selected trials. The spatiotemporal and kinematic gait parameters were compared between the groups during UGT. Results: In phase one, the fast group had a significantly greater angle and angular velocity of knee flexion and ankle plantar flexion than the slow group (*p* < 0.05). Phase two showed that the fast group had a significantly greater angle and angular velocity of knee extension than the slow group (*p* < 0.05). Concerning the correlation analysis, the angle and angular velocity of knee flexion and ankle plantar flexion showed a negative correlation with the SSRT during UGT in phase one (*p* < 0.05). Phase two showed that the angle and angular velocity of knee extension was negatively correlated with the SSRT during UGT (*p* < 0.05). Conclusion: The shorter the SSRT, the greater the angle and joint angular velocity of the ankle or knee joint that were prepared and adjusted for gait termination. The correlation between the SSRT and UGT suggests that a participant’s capacity to inhibit an incipient finger response is associated with their ability to make a corrective gait pattern in a choice-demanding environment.

## 1. Introduction

Response inhibition is a key component of executive function that supports behavioral flexibility by stopping highly automated inappropriate actions [1]. Response inhibition is assessed using a stop-signal task (SST), which requires the individual to suppress the motor response upon presentation of an external stimulus or cue [2,3]. Response inhibition deficits have been observed in patients with Parkinson’s disease [4] and individuals with mild cognitive impairment [5]. Loss of response inhibition is associated with severity of motor symptoms and freezing of gait, which are particularly debilitating components of the disease [6,7].

Stop-signal task is a well-established method for evaluating response inhibition because it specifically assesses an individual’s capacity to suppress an ongoing or already initiated reaction in response to a stop signal [3,8]. This stochastic model provides theoretically supported estimates of stopping latency [i.e., stop-signal reaction time (SSRT)]. This estimation is important because of the unobservable latency in the stopping process. The SSRT is a useful indicator of cognitive processing mechanisms associated with stopping [3,9]. Cognitive neuroscientists use the SSRT criterion to investigate whether it involves explicitly neural mechanisms in response inhibition [9,10]. Inhibitory control has recently been reported to play a role in fall prevention by suppressing potentially dangerous postural responses [11]. Rydalch et al. suggested that individuals with shorter SSRT showed less activation in the stepping leg when an obstacle was present compared to the trials that allowed for a recovery step [12]. This indicates a correlation between the capacity for response inhibition during a compensatory stepping action and their stopping ability, as measured by SST. It is intriguing that a measure derived from seated participants, responding to focal finger movements, generalizes performance on a whole-body balance recovery task [12,13]. Additionally, unexpected activities, such as the SST, disrupts action and impact cognition by imposing a global suppressive effect from fronto-basal ganglia network activation [14].

Gait termination is a transient behavior that necessitates complete forward momentum deceleration and a transition to a stable posture [15]. Gait termination occurs naturally in everyday life under various planned and unplanned environmental conditions. Planned gait termination is predetermined and occurs at the desired location, owing to interaction with environmental constraints. Unplanned gait termination occurs because of unexpected external stimuli [16]. The risk of falling increases when older adults are required to generate a postural response quickly and with an unplanned motor response [17]. Therefore, it is more challenging for the postural control mechanism to stop with unexpected interference than to overcome a low-height obstacle or turn around to avoid it [18]. An individual should either stop or change direction while walking to avoid collisions with unexpected environments. Both methods reduce the forward momentum of the center of mass and should be controlled within the base of the support. This sudden shift in the center of mass reduces stability and increases the risk of falling [19,20]. Therefore, gait termination may necessitate rapid deceleration of the center of mass during unexpected events [21].

During imaginary tasks involving gait termination, activations were observed in the right prefrontal area, right inferior frontal gyrus, and pre-supplementary area [22]. These areas are involved in the response inhibition process [23,24,25]. Furthermore, locomotion and postural control activities may occur in this area [26]. Based on this neurological evidence, SST and gait termination appeared to be associated with spatiotemporal and kinematic parameters. Few studies have examined the association between SSRT and corrective balance response [12,13]; however, no study has addressed SSRT and gait termination. Therefore, this study aimed to investigate gait parameters during unplanned gait termination in two groups classified according to SSRT. Additionally, the study analyzed the correlation between gait parameters and SSRT.

## 2. Methods

### 2.1. Participants

Twenty healthy participants were recruited for the study. The inclusion criteria were as follows: (1) no history of neurological or psychiatric disease; (2) no orthopedic surgery during the previous one year; and (3) no visual, vestibular, or balance disorder. All the participants were right-handed, as verified by a handedness questionnaire using the modified Edinburgh Handedness Inventory. Prior to conducting this study, the participants were divided into fast and slow groups by ranking the upper and lower halves of the group using the SSRT measured via SST, according to the group classification method used in a previous study [12]. All the participants were informed of the purpose of the study, and they voluntarily signed a participation consent form. The study protocol was approved by the Institutional Review Board of the local university (DKU 2021-03-062). Demographic and clinical data were collected, and no significant differences were found between the groups.

### 2.2. Stop-Signal Task (SST)

The stop-signal task was implemented in the STOP-IT program in the 2015 version of De Leeuw [27]. The STOP-IT jsPsych is a GNU-licensed program that can be installed on local computers. The SST consisted of go (75% of all trials) and stopped (25%) trials. Each trial began with a fixation sign, followed by the go stimulus after 250 ms in both sections. The stimulus was displayed on the screen until the participant responded using predefined response keys or until 1250 ms (i.e., the maximal response time) had passed. The inter-stimulus interval was set to 750 ms, regardless of the reaction time. A stop signal was presented following a variable stop-signal delay (SSD) in stop trials. The SSD was initially set to 300 ms and was adjusted continuously using the staircase tracking procedure. When inhibition was successful, the SSD increased by 50 ms; when inhibition was unsuccessful, the SSD decreased by 50 ms. The SSD was varied to maintain a 50% probability of correctly inhibiting a go response (i.e., pressing a “←” or “→” key) following the stop signal presentation. The SSRT was calculated using the SSD, in which the participants were able to suppress their responses 50% of the time. In the go trials, the participants were required to respond quickly and accurately to these arrows (i.e., pressing a “←” or “→” key). In the stop-signal trials, the arrows turn red (i.e., a visual stop-signal) after a variable delay, instructing the participants to withdraw their responses. The STOP-IT data were analyzed using the ‘R Shiny’ application. The software adhered to the recommendations outlined in a previous study, and the stop-signal reaction time (SSRT) was calculated using the integration method with the replacement of go omissions [28]. The following variables were measured through SST: (1) SSD: the average delay between the presentation of the go stimulus and stop-signal; (2) SSRT: the average time it takes to stop a response; and (3) go reaction time (goRT): the reaction time in go trials, including choice errors.

### 2.3. Motion Capture System

Gait termination data were collected using a Qualisys motion analysis system with six cameras (Qualisys AB, Göteborg, Sweden) at a sampling rate of 100 Hz. Force data were collected using two Bertec force plates (AM6500, Bertec Corporation, OH, USA) at 1000 Hz. The two force plates were synchronized with a Qualisys motion analysis system (with AM6800 amplifiers) in the middle of the walkway. It covered the two force plates and surrounding area in gray carpets that matched the laboratory floor. The modified conventional gait model was used to determine the segment placement and movement [29]. Anatomical markers were placed bilaterally on the sacrum, anterior superior iliac spine, trochanter major, lateral aspect of the thigh, lateral femoral condyle, lateral aspect of the shank, lateral malleolus, 2nd metatarsal head, and calcaneus.

The motion for unplanned gait termination after the stop signal was divided into phases because the kinematics of each gait cycle were distinct. Three phases of gait have been identified based on previous studies analyzing distinguishable vertical ground reaction force landmarks [30,31]. The data for unplanned gait termination were divided into three phases: phase one (when the heel of the trailing foot contacted the force plate until GRF-z of the trailing foot was mid-stance valley), phase two (when GRF-z of the trailing foot was a mid-stance valley until the heel of the leading foot contacted the force plate), and phase three (when the heel of the leading foot contacted the force plate until the participant’s anteroposterior center of mass velocity fell below 90% measure) [32]. Unplanned gait termination trials were analyzed for each participant, excluding unnatural trials in which the participants took an extra step after failing to stop or appeared to predict the stopping point and slowed their walking speed. Data from all the participants were averaged over the three trials. The spatiotemporal gait parameters [gait termination time (GTT), GTT/velocity, step length, step length/height, and gait velocity) and kinematic gait parameters (joint angle and joint angular velocity) were analyzed using a tool within Qualisys. Kinematic parameters of the leading limb were calculated for phases one, two, and three of the gait cycle, as primary outcome measures. The GTT and step length were normalized for the gait velocity, height, and body mass of each participant.

### 2.4. Experimental Procedure

The SST program began before the unplanned gait termination trial. All the participants were seated in front of a table with their left hand on the response key, and the SST was performed. After filling in their age and sex, the experiment entered full-screen mode. SST instructions were displayed on the screen. The SST procedure was divided into two sessions: practice (one block of 32 trials) and experimental (four blocks of 64 trials) sessions. All the participants completed the SST sessions three times.

Before the unplanned gait termination trials, the participants were asked to walk sufficiently on a walkway without controlling their walking pace to induce natural motions. The participants were asked to complete the test at their own pace, and instructed to stop suddenly, as soon as they heard a stop signal. The participants were not told which force plate to stop before the unplanned gait termination trials. They were instructed to freeze immediately upon hearing the stop signal and maintain their position until they were instructed to move again. The stop signal emerged when the dominant leg contacted the first force plate. The stop signal was configured using Bertec Acquire 4 to generate a sound when GRF-z increased by 50 N or more [33]. The participants were provided with a practice trial in which the auditory cue was provided to ensure a proper understanding of the task. The ‘walk-through’ and ‘gait termination’ were respectively determined by setting the ratio to 75% and 25%. At least three trials were conducted when the leading foot landed on a force plate. To eliminate the learning effect, stopping points and attempts to stop walking were varied. Only the data collected and analyzed when both feet were successfully placed on the force plates were used to measure the biomechanical parameters.

### 2.5. Statistical Analysis

Data analysis was performed using SPSS Statistics for Windows (version 20.0; SPSS Inc., Chicago, IL, USA). The Shapiro–Wilk test was used to determine the normal distribution of each group. The Mann–Whitney U and chi-squared tests were used to compare differences in sex, age, height, and weight between the groups. The Mann–Whitney U test was used to compare spatiotemporal gait parameters (GTT, GTT/velocity, step length, step length/height, and gait velocity) and kinematic gait parameters (joint angle and joint angular velocity). The Spearman’s rank correlation coefficient was used to analyze the correlation between the gait parameters and SSRT. Null hypotheses of no difference were rejected if the *p*-values were less than 0.05. The results are presented as mean ± standard deviation.

## 3. Results

No significant differences were observed in demographic data between the groups in terms of sex, age, height, and weight (*p* > 0.05) (Table 1). Table 1 lists the temporal parameters during the SST for each group. The fast group was significantly greater in the SSD group and lower in the SSRT than slow group (*p* < 0.05), and there were no significant differences in goRT between the groups (*p* > 0.05).

In terms of all spatiotemporal gait parameters during unplanned gait termination, there were no significant differences between the groups (*p* > 0.05) (Table 2). Table 3 shows a comparison of kinematic parameters during unplanned gait termination in the phases between the groups. In phase one, the fast group had a significantly greater angle and angular velocity of knee flexion and ankle plantar flexion than the slow group (*p* < 0.05). There was no significant difference in the angle and angular velocity of the hip flexion between the groups (*p* > 0.05). Phase two showed that the fast group had a significantly greater angle and angular velocity of knee extension than the slow group (*p* < 0.05), but there were no significant differences in the angle and angular velocity of the hip and ankle joints between the groups (*p* > 0.05). In phase three, there was no significant difference between the groups in any of the kinematic gait parameters (*p* > 0.05) (Figure 1).

Concerning the correlation analysis (Figure 2 and Figure 3), in phase one, the angle of knee flexion and ankle plantar flexion showed a negative correlation with the SSRT during unplanned gait termination (*p* < 0.05). The angle of hip flexion was not significantly correlated with SSRT (*p* > 0.05). Phase two showed that the angle of knee extension was negatively correlated with the SSRT during unplanned gait termination (*p* < 0.05). However, the angle of hip extension and ankle plantar flexion was not significantly correlated with SSRT (*p* > 0.05). In phase 3, there was no significant correlation between any of the angular parameters and SSRT (*p* > 0.05). According to the results of comparing the angular velocities between the groups, phase one showed that the angular velocity of knee flexion (*p* < 0.05) and ankle plantar flexion (*p* < 0.05) was negatively correlated with the SSRT during unplanned gait termination. The angular velocity of hip flexion was not significantly correlated with SSRT (*p* > 0.05). In phase two, the angular velocity of knee extension was negatively correlated with the SSRT during unplanned gait termination (*p* < 0.05), but there was no significant difference between SSRT and angular velocities of the hip and ankle joints (*p* > 0.05). In phase 3, there was no significant correlation between any of the angular velocity parameters and SSRT (*p* > 0.05).

## 4. Discussion

The purpose of this study was to determine whether response inhibition, as measured by SST, was associated with unplanned gait termination. Participants with a shorter SSRT revealed a greater joint angle and angular velocity of the ankle or knee joint in preparation for gait termination. This indicates that response inhibition, as measured by the SST, is associated with a participant’s ability to suppress behavior during gait termination. The fact that response inhibition derived from finger responses using SST affects whole-body performance, such as unplanned gait termination, is of particular interest.

The SSRT was significantly lower in the fast group than in the slow group. Rydalch et al. used SSRT to rank the upper and lower halves to investigate the correlation between SSRT and compensatory balance responses. Based on this study, we divided the participants into fast and slow groups [12]. Additionally, there was no significant difference in goRTs in this study, which suggests that participants did not react strategically to perform response inhibition during the SST [3].

A previous study investigated the magnitude of the effect of gait velocity on gait biomechanics according to age [34]. In young adults, gait parameters indicated large effect sizes for cadence, step length, and stride length at slower speeds. The cadence and step length had large effect sizes at shorter speeds, indicating that these parameters increased as the velocity increased [32]. Therefore, it was reported that the gait velocity could affect the success rate of gait termination, GTT, and step length under the condition of unplanned gait termination [35]. However, in the present study, there were no significant differences in spatiotemporal gait parameters between the groups. The results of this study showed that the factors that could affect spatiotemporal gait parameters (GTT and step length) were minimized.

This study analyzed unplanned gait termination by dividing the gait cycle based on the leading limb. In phase one, the fast group showed that the angles of knee and ankle plantar flexions were significantly greater, and the angular velocity of knee and ankle plantar flexions was significantly greater than that of the slow group. During the initial swing, the hip, knee, and ankle are flexed to advance the limb forward and create foot clearance over the ground, in which the foot is pushed and lifted off the ground [33]. For safe gait termination, forward movement of the body must be slowed down to stabilize the upright position [36,37]. The abilities required to maintain stability, weight transfer, and foot clearance become more important during these transition phases than during steady-state gait conditions [38,39]. Therefore, the fast group was thought to have quickly changed the movement pattern for gait termination immediately after the stop signal. Furthermore, neural networks that contribute to cognitive response inhibition can also contribute to unplanned gait termination, given the general characteristics of stopping [14]. However, there was no significant difference in the angle and angular velocity of hip flexion between the groups. It is thought that it did not show greater movement due to the relatively short GTT and anatomical characteristics of the hip joint [40].

In phase two, the fast group showed a significantly greater angle and angular velocity of knee extension than did the slow group. There were no significant differences in the angular velocities of hip extension and ankle plantar flexion and the angles of hip extension and ankle plantar flexion between the groups. Previous studies have reported that hip and knee strategies are used during gait termination [41,42]. Effective gait termination necessitates greater hip and knee joint flexion followed by hip and knee extension, indicating a well-defined pattern of flexion–extension movements in the leading limb [41,42]. In the present study, the participants flexed the knee joint more and then extended it after the stop signal in phase two. We considered that the knee strategy could be adopted as a supplementary means for rapid gait termination in the same hip and ankle joint strategy.

There was no significant difference in the kinematic parameters between the groups in phase three. Previous studies on gait termination were mainly analyzed based on the condition or disease. For instance, they performed gait analysis according to conditions, such as planned or unplanned gait termination [41,43,44,45,46]. Previous studies have investigated gait termination strategies in patients with musculoskeletal and neurological disorders. Moreover, unplanned gait termination was compared in patients with musculoskeletal or neurological disorders and controls [40,46,47]. When analyzing unplanned gait termination in normal participants, gait analysis was performed according to the walking speed [43]. Based on these findings, we assumed that this study was conducted on normal participants who could perform unplanned gait termination and that it was conducted under the same experimental conditions in phase three.

The correlation analysis showed that SSRT had a negative correlation with the angle and angular velocity of knee and ankle plantar flexions in phase one. The SSRT showed a negative correlation with the angle and angular velocity of knee extension in phase two. However, there was no significant correlation between the kinematic gait parameters and SSRT in phase three. These results suggest that the response inhibition measured in the cognitive task was associated with the gait pattern for unplanned gait termination, which should suppress the response to intermittent forward movement. Such a relationship would support a shared underlying cognitive mechanism in both behavioral responses [14].

The present study investigated the association between SSRT and unplanned gait termination, and it may seem logical that the overall neural networks involved in motor inhibition were similar. However, this study has several limitations. First, we conducted this study with normal adults; therefore, it was difficult to generalize the findings for all individuals. Further studies are needed to determine the association between gait termination and patients with Parkinson’s disease or older adults who lack cognitive response inhibition. Second, this study used only kinematic parameters as outcome measures. In future studies, it will be necessary to analyze the electromyography and ground reaction force during unplanned gait termination. Third, this study compared the temporal parameters measured by SST and the overall biomechanical parameters of gait termination; therefore, it was difficult to analyze the neurological association between the tasks. Future studies are needed to determine the relationship between SSRT and gait termination through neurological analysis.

In conclusion, this study investigated differences in gait parameters for unplanned gait termination between groups classified using the SSRT. The shorter the SSRT, the greater the angle and joint angular velocity of the ankle or knee joint that were prepared and adjusted for gait termination. The correlation between the SSRT and unplanned gait termination suggests that a participant’s capacity to inhibit an incipient finger response is associated with their ability to make a corrective gait pattern in a choice-demanding environment. Moreover, this study provides basic knowledge to comprehensively investigate cognitive and motor inhibition during unexpected events. We suggest that the findings of this study indicate that aging-related declines in cognitive response inhibition can increase the risk of falls due to a decrease in physical inhibition ability.

## Figures and Tables

**Figure 1 brainsci-13-00304-f001:**
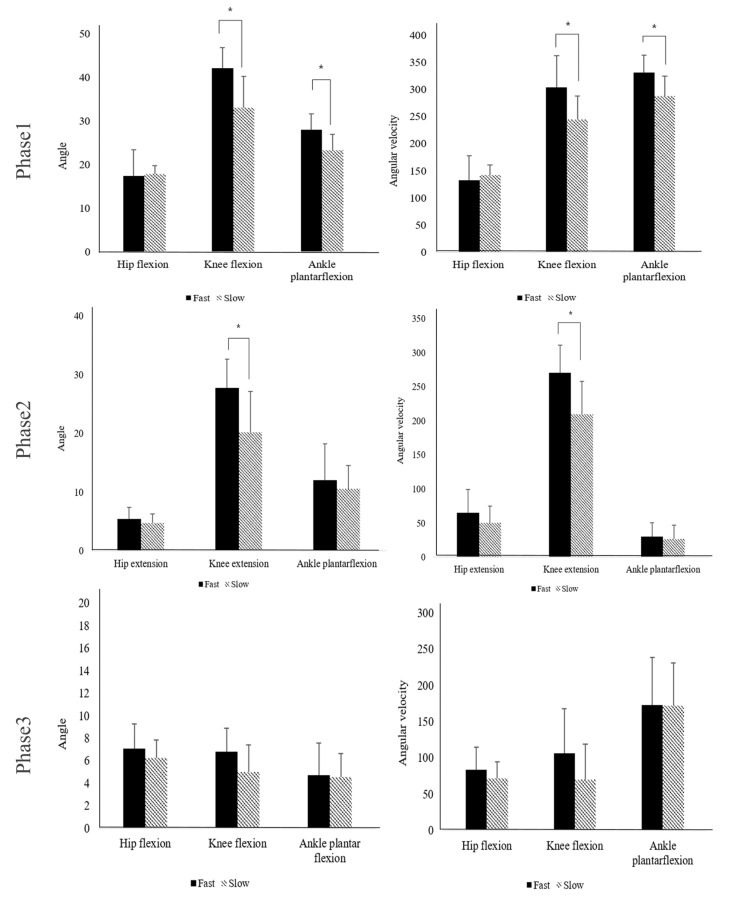
Comparison of joint angle and joint angular velocity between the groups in all phases. * Significant difference in the joint angle between the groups (*p* < 0.05).

**Figure 2 brainsci-13-00304-f002:**
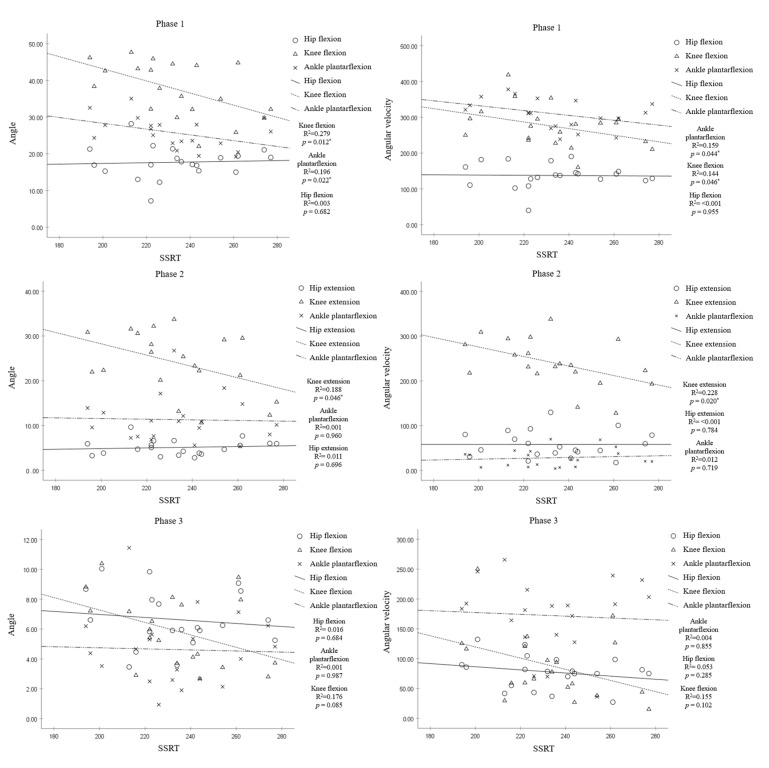
Correlation between SSRT and joint angle and joint angular velocity in all phases. * Significant correlation between SSRT and joint angle or joint angular velocity (*p* < 0.05).

**Figure 3 brainsci-13-00304-f003:**
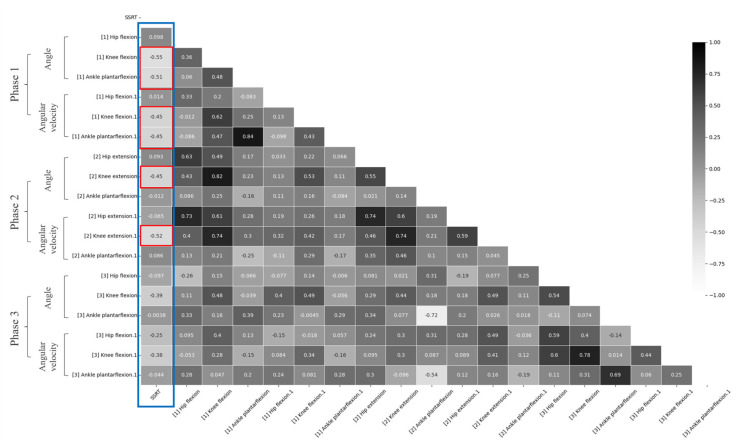
A heatmap depicting the correlation between SSRT and gait parameters for each phase; Blue box: correlation analysis between SSRT and gait parameters, Red box: gait parameters showing significant correlation with SSRT.

**Table 1 brainsci-13-00304-t001:** General characteristics and comparison of stop-signal task parameters between the groups.

	Fast-Group	Slow-Group	x^2^/Z	*p*-Value
Sex (M/F)	4/6	5/5	−0.438	0.653
Age (years)	26.40 (4.70)	25.50 (3.24)	−0.726	0.468
Height (cm)	169.50 (9.41)	168.90 (4.31)	−0.038	0.970
Weight (kg)	63.00 (10.38)	65.80 (16.55)	−0.416	0.677
SSD (ms)	192.10 (64.61)	125.70 (55.45)	−2.117	0.034 *
SSRT (ms)	214.50 (13.22)	252.60 (15.41)	3.781	<0.001 *
goRT (ms)	418.10 (65.46)	383.40 (67.92)	1.285	0.199

Values represent mean (±standard deviation), M: male, F: female, SSD: stop-signal delay, SSRT: stop-signal reaction time, goRT: go reaction time, * Significant difference between the groups (*p* < 0.05).

**Table 2 brainsci-13-00304-t002:** Comparison of kinematic parameters during the unplanned gait termination in all phases between the groups.

	Fast-Group	Slow-Group	Z	*p*-Value
**Phase 1**				
*Joint angle*				
Hip flexion (°)	17.47 (6.01)	17.93 (1.88)	−0.227	0.821
Knee flexion (°)	42.12 (4.71)	33.12 (7.19)	−2.646	0.008 *
Ankle plantar flexion (°)	27.98 (3.70)	23.36 (3.60)	−2.343	0.019 *
*Angular velocity*				
Hip flexion (°/s)	132.65 (45.44)	142.39 (18.78)	−0.680	0.496
Knee flexion (°/s)	303.98 (58.71)	244.65 (43.12)	−2.343	0.019 *
Ankle plantar flexion (°/s)	331.29 (32.69)	287.83 (37.44)	−2.268	0.023 *
**Phase 2**				
*Joint angle*				
Hip extension (°)	5.43 (1.96)	4.77 (1.50)	−0.756	0.450
Knee extension (°)	27.77 (4.84)	20.21 (6.96)	−2.343	0.019 *
Ankle plantar flexion (°)	12.05 (6.17)	10.57 (3.97)	−0.302	0.762
*Angular velocity*				
Hip extension (°/s)	65.52 (33.56)	50.59 (24.27)	−1.058	0.290
Knee extension (°/s)	270.44 (40.87)	209.84 (48.34)	−2.343	0.019 *
Ankle plantar flexion (°/s)	30.03 (20.25)	26.49 (20.82)	−0.529	0.597
**Phase 3**				
*Joint angle*				
Hip flexion (°)	7.05 (2.19)	6.24 (1.60)	−0.756	0.450
Knee flexion (°)	6.78 (2.09)	4.98 (2.42)	−1.663	0.096
Ankle plantar flexion (°)	4.71 (2.87)	4.53 (2.09)	−0.151	0.880
*Angular velocity*				
Hip flexion (°/s)	83.63 (31.07)	71.58 (23.12)	−1.209	0.226
Knee flexion (°/s)	106.01 (61.81)	70.37 (48.52)	−1.587	0.112
Ankle plantar flexion (°/s)	172.53 (65.66)	171.77 (59.14)	0.000	1.000

Values represent mean ± standard deviation, (°): range of motion, (°/s): angular velocity, * Significant difference between the groups (*p* < 0.05).

**Table 3 brainsci-13-00304-t003:** Comparison of spatiotemporal parameters during the unplanned gait termination between the groups.

	Fast Group	Slow Group	Z	*p*-Value
GTT (sec)	1.05 (0.14)	1.12 (0.80)	−0.945	0.345
GTT/v (s^2^/m)	0.08 (0.01)	0.09 (0.01)	−1.134	0.257
Step length (cm)	54.61 (7.70)	55.77 (4.43)	−0.302	0.762
Step length/h (ratio)	0.32 (0.04)	0.33 (0.03)	−0.454	0.650
Gait velocity (m/s)	1.26 (0.09)	1.23 (0.09)	−0.832	0.406

Values represent mean (±standard deviation, GTT: gait termination time, GTT/v: gait termination time/gait velocity, step length/h: step length/height.

## Data Availability

The data presented in this study are available on request from the corresponding author.

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
