# Peer review of "Biomechanical Analysis of Unplanned Gait Termination According to a Stop-Signal Task Performance: A Preliminary Study"

_brainsci, 2023, doi:10.3390/brainsci13020304_

Round 1

Reviewer 1 Report

_The purpose of this study was to investigate the gait parameters of the lower extremity that occurred during unplanned gait termination (UGT) in two groups classified by the stop signal reaction time (SSRT).

_Gait is classified into three phases, and for each of these, measurements and discussion are made for the fast and slow SSRT groups.

_The faster the SSRT, the greater the angle and joint angular velocity of the ankle or knee joint that were prepared and adjusted for gait termin ation.

In general, Gait Analysis divides gait into Stance Phase and Floating Phase, and classifies each of them in detail according to the geometric posture situation. In this paper, the division into three Phases is not explained, but the validity of this division is not explained well, which is an important point because reaction time and posture changes are considered to be greatly affected by Gait Phase.

When the brain commands a change in behavior pattern during a person's movement, the muscles first receive the command value and an action to change the movement occurs. In conjunction with this, torque is generated to change the posture of the joints. Considering that the change in torque is expressed in terms of the angular acceleration of the joint, the difference in reaction speed is considered to affect the difference in angular acceleration. The joint velocity is considered to be a change to converge to zero joint velocity, which is the state of gait stop, and is considered to be dependent on the angular acceleration and also influenced by the state of the Gait Phase. On the other hand, the joint angle is only a result of the change to converge to zero joint speed, which is the state of gait standstill, and is considered to be more influenced by the state of Gait Phase, Gait Speed, and the subject's body size. I think that to show clear correlation between the SSRT and unplanned gait termination, more evidence is necessary. It would be easier to understand by displaying the results visually.

Author Response

Thank you for your detailed review. Please see the attachment.

Reviewer 2 Report

General questions

Congratulations for the nice work. This work presents a good scientific relevance. However, some issues need to be clarified in addition to other small corrections.

Major questions

- I suggest using line numbering to facilitate text review

- Table 1: the fast group should not have less stop-signal delay?

- It remains to present correlation results in phase 3 (page 5), even if it was not significant

- the study was carried out with young and healthy people. How might these results be useful for other populations (diseases for example)?

- I advise not to relate speed with SSRT (the authors use the term faster SSRT). In reality SSRT is shorter because it is a temporal measure. Despite representing a faster response, SSRT is actually shorter. This involves revising the text and making adaptations.

- throughout the text and mainly in the methods it is not clear that the SSRT was evaluated through a test using the fingers. Make it clearer, especially in the text referring to SST.

Minor questions

- Please, to see number 1 at the end of the phrase: “This sudden shift in the center of mass reduces stability and increases the risk of falling [19,20]1.

- Is this sentence correct? is confused. “The fast group was significantly greater in the SSD group and lower in the SSRT group than in the slow group...”

- references: ...disorders in P arkinson's disease: A data‐driven... (space in P arkinson’s)

Author Response

(The authors gave the same response as above.)

Round 2

Reviewer 2 Report

Congratulations for the nice work

Author Response

Thank you for your detailed review to Reviewer 2.

Also, I am attaching an answer to deliver to the Academic Editor.
